# Comparative Analysis of Gut Microbial Community Structure of Three Tropical Sea Cucumber Species

Yanan Wang [1,2], Yue Zhang [1,2], Chenghao Jia [2,3], Qiang Xu [1,2], Yun Rong [1,2], Zening Xu [1,2], Yuanhang Wang [1,2] and Fei Gao [1,2,*]

1    School of Marine Biology and Aquaculture, Hainan University, Haikou 570228, China; 20070703210009@hainanu.edu.cn (Y.W.); rella510@126.com (Y.Z.); xuqianghnu@hainanu.edu.cn (Q.X.); 18668072297@139.com (Y.R.); 2122095134060@hainanu.edu.cn (Z.X.); 2121090800035@hainanu.edu.cn (Y.W.)

2    State Key Laboratory of Marine Resource Utilization in South China Sea, Hainan University, Haikou 570228, China; xicheng121@yeah.net

3    School of Ecology and Environment, Hainan University, Haikou 570228, China

*    Correspondence: gaofeicas@126.com

**Abstract:** Based on 16S rRNA gene high-throughput sequencing technology, the microbial community structure in the gut of three tropical sea cucumber species, *Holothuria atra*, *Stichopus chloronotus*, and *S. monotuberculatus*, and their habitat sediments were analyzed. The OTUs in the gut contents of *S. monotuberculatus*, *S. chloronotus*, *H. atra*, and their sediments were 2489 ± 447, 1912 ± 235, 1628 ± 150, and 4162 ± 94, respectively. According to alpha diversity analysis (Ace, Chao1, Shannon, Simpson), the richness and diversity of microflora in the gut of all three sea cucumber species were significantly lower than that in sediments ($p < 0.01$). Anosim analysis showed that the intra-group differences were less than the inter-group differences (R > 0), and the microbial community composition of the sediment was distinctly different from those of three sea cucumber species ($p < 0.05$). UPGMA tree and Anosim analysis also revealed that the gut microbial communities of *H. atra* and *S. chloronotus* were more similar than that of *S. monotuberculatus*. Proteobacteria was the predominant phylum in all samples, and there was no significant difference in relative abundance among all groups. Actinobacteria was also a dominant phylum, and the relative abundance in *S. chloronotus* was significantly higher than that in other samples ($p < 0.05$). Potential probiotics and sequences related to *Lactobacillus* and *Pseudomonas* that may be developed for sea cucumber culture were also found in the study. It is speculated that the main reason for the difference in microbial communities between gut microbiota and environmental sediments may be the unique and quite different environment in the digestive tract of sea cucumbers. Whereas, the differences in gut microbiota among the three sea cucumber species were caused by selective feeding. These findings may provide basic data for tropical sea cucumber gut microflora studies and assist in the sea ranching and aquaculture development of the tropical sea cucumber species.

**Keywords:** sea cucumber; bacteria; gut; tropical; 16S rRNA

## 1. Introduction

Sea cucumber (1775 species) represents marine invertebrate species that constitute the Holothuroidea class within the phylum Echinodermata [1]. They ingest sediments to extract the organic constituents or feed upon particles suspended in the water column, playing a significant role in nutrient cycling and sediment modification in shallow-water habitats [2,3]. In addition, some sea cucumbers are of great economic value because they are rich in nutrients, and many bioactive compounds can be extracted for use in the medical and pharmaceutical industries [4–7]. Due to the long-term overfishing and degradation of coral reefs, tropical sea cucumber resources are in continuous decline, and endangered germplasm resources need to be restored and protected.

The gut microbiota of animals is involved in a multitude of important physiological processes of the host, including nutrition absorption, metabolism, growth and development, immunity, etc. [8–11]. In echinoderm sea cucumber, recent studies proved that the gut microbiota provides nutrition for the host and participates in the host's energy metabolism and immune defenses. Firstly, the bacteria ingested from the surface sediment are a commonly reported component of holothuroid diets [12], and Moriarty [13] showed that the sea cucumber species *Holothuria atra* and *Stichopus chloronotus* had higher efficiency at assimilating bacteria than organic carbon, averaging 30–40%. The holothurians can indirectly use the bacteria to provide them with the otherwise not available essential nutrients, such as amino acids, vitamins, and trace elements [14–16]. Moreover, gut microbes play an important role in digestion, and they can secrete a variety of digestive enzymes, such as amylase, protease, cellulase, phosphatase, and lipase, and are involved in the digestion of detritus, lipids, and various polysaccharide degradation activities [17–19]. Additionally, the gut bacteria have the function of immunity for hosts, and the probiotic bacteria from the gut of *Apostichopus japonicus* have positive effects on the disease resistance of juvenile *A. japonicus*, especially in significantly improving the disease resistance to *Vibrio splendidus* exposure [20].

Holothurians *H. atra*, *S. chloronotus*, and *S. monotuberculatus* are all large-deposit-feeding species with significant ecological and economic values, abundantly existing in the Indian and Pacific Oceans [2,3,21,22]. Despite the multiple functions of gut microflora for hosts, there are only a few reports on the gut microbial communities of *H. atra* and *S. chloronotus*, which mainly focused on culturable bacteria. Moriarty [13] determined the culturable bacterial biomass by muramic acid measurements in the sediments and gut contents of *H. atra* and *S. chloronotus* on the Great Barrier Reef. Ward-Rainey et al. [21] investigated the amounts of culturable aerobic bacteria in the digestive tract of *H. atra* and the surrounding sediment by 16S ribosomal DNA sequence analysis and concluded that the number of aerobic bacteria in the sediment was higher than that in the intestine. So far, there is no relevant research about the gut microbiota in *S. monotuberculatus*. Since most bacteria in nature are not culturable under laboratory conditions, only a small fraction of the bacterial community in the gut of the three tropical sea cucumbers has been revealed.

In recent years, with the sequencing methods continuing to evolve, to overcome the nonculturable and genomic diversity of most bacteria, high-throughput sequencing technology has been widely used in the analysis of the gut microbial community structure of aquatic animals, such as shrimps, *Penaeus japonicus* and *Penaeus monodon*, and Antarctic fish, *Trematomus bernacchii*, *Chionodraco hamatus*, *Gymnodraco acuticeps*, and *Pagothenia borchgrevinki* [23–26]. Studies on the gut microbial community of the sea cucumber *A. japonicus* based on high-throughput sequencing technology have been reported [27–29]. In summary, the purpose of this study is to investigate whether the gut microbiota composition of the three species of sea cucumber differ in the same natural environment. This study investigated the microbial community composition in the guts of three common tropical sea cucumbers, *S. monotuberculatus*, *S. chloronotus*, and *H. atra*, and their surrounding surface sediments using 16S rRNA gene high-throughput technology. The study of gut microbiota in tropical sea cucumbers is still in the early stages of research, and this study may provide basic data for the study of intestinal microbiota in tropical sea cucumbers. Previous studies have found differences in habitat and food selection among three species of sea cucumbers [3,22,30,31]. By comparing the structures of the microbial communities in the surrounding sediments and the gut microbiota of three sea cucumber species, the feeding preferences of sea cucumbers could be understood from another perspective, which may help to reveal the relationship between their microflora in the gut and environmental habitats and to supplement the current research on gut microorganisms of tropical sea cucumbers.

## 2. Materials and Methods

### 2.1. Sample Collection

In this study, all sea cucumber and sediment samples were collected in the northern sea area of Wuzhizhou Island, Sanya, Hainan, China (18°18′58″ N, 109°45′41″ E), in June 2020. The seawater temperature at the time of sample collection was 27.44 °C, the salinity was 33.4, and the water depth was about 9–14 m. As these sea cucumbers lived in a natural habitat, they fed only on natural diets. The sea cucumber species were: *S. monotuberculatus* (SM, N = 5), *S. chloronotus* (SC, N = 5), and *H. atra* (HA, N = 5). Our previous surveys showed that *S. monotuberculatus* usually live under rocks in the cracks and crevices of reefs, yet *S. chloronotus* and *H. atra* live in the surrounding sandy-bottom area. During the sampling process, based on the principle of community ecotone, we hypothesized that the sediments at the intersection of the two habitats shared the characteristics of sediment from both habitats. As a result, the surface sediments (SS, N = 5) we collected were located at the confluence of the two habitats, collected using the 50 mL syringe (the diameter of the adjusted syringe opening was 2.9 cm, and the collection depth was less than 1 cm) [32,33]. Afterward, the sea cucumber and sediment samples were stored in ice boxes and promptly transported to the laboratory within an hour.

Before dissection, the surface skin of the sea cucumbers was sterilized with 75% ethanol to reduce exogenous bacterial contamination. Under sterile conditions, the ventral surface was dissected with a sterile scalpel to expose the gut in the body cavity. The gut contents were squeezed out from the digestive tract (foregut 2–3 cm) and collected in a sterile cryotube. The gut content samples and the ambient sediment samples were stored at −80 °C for further analysis.

### 2.2. DNA Extraction and PCR Amplification

According to the manufacturer's protocol, DNA was extracted from gut content samples and sediment samples using the Soil DNA Kit (Omega Biotech, Norcross, GA, USA) in the laboratory.

An appropriate amount of DNA was taken into the centrifuge tube and diluted to 1 ng/μL with sterile water. PCR amplification of bacterial 16S rRNA gene hypervariable regions (V3–V4) was conducted using the universal primers, including 341F (5′-CCTAYGGGRBGCASCAG-3′) and 806R (5′-GGACTACNNGGGTATCTAAT-3′).

The total 30 μL of the PCR mixture contained 15 μL of Phusion $^{®}$ High-Fidelity PCR Master Mix (New England Biolabs, Ipswich, MA, USA), 3 μL of forward and reverse primers, 2 μL of $H_2O$, and 10 ng of template DNA. Thermal cycling consisted of initial denaturation at 98 °C for 1 min, followed by 30 cycles of denaturation at 98 °C for 10 s, annealing at 50 °C for 30 s, elongation at 72 °C for 30 s, and a final extension step at 72 °C for 5 min.

### 2.3. High-Throughput Sequencing

According to the manufacturer's protocol, sequencing libraries were generated using the TruSeq$^{®}$ DNA PCR-Free Sample Preparation Kit (Illumina, San Diego, CA, USA). The library concentration was assessed on the Qubit@ 2.0 Fluorometer (Thermo Scientific, Carlsbad, CA, USA) system. Finally, the library was sequenced by the NovaSeq6000 platform and 250 bp paired-end reads. All data were sequenced by Novogene (Tianjin, China).

### 2.4. Data Analysis

Quality filtering on the raw reads was performed under specific filtering conditions to obtain the high-quality clean reads according to the Cutadapt quality-controlled process [34] (V1.9.1, http://cutadapt.readthedocs.io/en/stable/ (accessed on 3 September 2020)). Paired-end reads from the original DNA fragments were merged using FLASH [35] (V1.2.7, http://ccb.jhu.edu/software/FLASH/ (accessed on 3 September 2020)). To obtain the clean reads, we used QIIME (V1.9.1, http://qiime.org/scripts/split_libraries_fastq.html

(accessed on 3 September 2020)) [36] to finish the reads' quality-control process and filter out the reads that had a continuous high-quality base length of less than 75% of the reads' length. An algorithm was used to detect chimera sequences (https://github.com/torognes/vsearch/ (accessed on 3 September 2020)), and then the chimera sequences were removed [37]. Finally, clean reads were obtained.

The sequences with ≥97% similarity were assigned to the same operational taxonomic units (OTUs). The representative sequence for each OTU was screened for further annotation. Species annotation was performed on the OTU sequence, and the Mothur method and the SSUrRNA database [38] of SILVA132 [39] were used for species annotation analysis (with a threshold of 0.8~1). Finally, the sample with the least amount of data was used as the standard for normalization. The subsequent alpha diversity analysis (to analyze the diversity of microbial communities within the sample) and beta diversity analysis (to compare the microbial community structure of different samples) were based on data after normalization.

To calculate the alpha diversity, we rarified the OTU table and calculated the following metrics: Observed species, Chao—the Chao1 estimator (http://scikit-bio.org/docs/latest/generated/skbio.diversity.alpha.chao1.html#skbio.diversity.alpha.chao1 (accessed on 10 September 2020)), Simpson—the Simpson index (http://scikit-bio.org/docs/latest/generated/skbio.diversity.alpha.html#skbio.diversity.alpha.simpson (accessed on 10 September 2020)), Shannon—the Shannon index (http://scikit-bio.org/docs/latest/generated/skbio.diversity.alpha.shannon.html#skbio.diversity.alpha.shannon (accessed on 10 September 2020)), and ACE—the ACE estimator (http://scikit-bio.org/docs/latest/generated/skbio.diversity.alpha.ace.html#skbio.diversity.alpha.ace (accessed on 10 September 2020)). For coverage, we used the Good's coverage (http://scikit-bio.org/docs/latest/generated/skbio.diversity.alpha.goods_coverage.html#skbio.diversity.alpha.goods_coverage (accessed on 10 September 2020)). All the indices in our samples were calculated with QIIME (Version1.7.0) and displayed with Rsoftware (Version 2.15.3, including packages ggplot2, ggpubr, ggsignif, ggprism, picante, dplyr, and RColorRrewer).

To analyze the differences between sample groups, we used QIIME software (Version 1.9.1) to calculate the Unifrac distance and construct a UPGMA sample clustering tree. We used R software (Version 2.15.3) to create principal component analysis (PCA) and principal coordinate analysis (PCoA) plots. WGCNA (weighted gene co-expression network analysis), stats, and ggplot2 packages of R software were used for PCoA and PCA. Anosim (analysis of similarities) uses the Anosim function of the R Vegan package. The R-value is between (−1, 1), if greater than 0, this indicates significant differences between groups. If the R-value is less than 0, this indicates that the intra-group difference is greater than the inter-group difference. The reliability of the statistical analysis is represented by a $p$-value, where $p < 0.05$ indicates statistical significance.

The bacterial function was predicted by Tax4Fun. It was achieved by the nearest-neighbor method based on minimum 16S rRNA sequence similarity. The 16S rRNA gene sequence of prokaryotes was extracted from the KEGG database and compared to the SILVA SSU Ref NR database via the BLASTN algorithm (BLAST bitscore > 1500). The functional information of prokaryotes from the KEGG database annotated by UProC and PAUDA was translated into the SILVA database to realize functional annotation in the SILVA database. OTUs were clustered based on SILVA database sequences, and then functional annotation information was obtained.

To find the differences between groups at different levels, an independent samples $t$-test was performed with the R software. Finally, in order to avoid the occurrence of "Type I error", we corrected the $p$-value to a q-value via the Benjamini and Hochberg (BH) method, as follows: (1) the $p$-values of each gene were ranked from the smallest to the largest, (2) the largest $p$-value remained as is, (3) the second largest $p$-value was multiplied by the total number of genes in a gene list divided, by its rank: if it was less than 0.05, it was considered significant: q-value = $p$-value × (n/n − 1), and (4) the third $p$-value was multiplied as in step 3: q-value = $p$-value × (n/n − 2), (5) and so on [40,41]. The statistical significance was set at 0.05. All values were expressed as mean ± SD.

## 3. Results

### 3.1. Sequencing Data Statistics and Analysis

A total of 1,302,978 clean reads were obtained from the 15 gut content samples from *S. monotuberculatus* (SM), *S. chloronotus* (SC), and *H. atra* (HA), and the 5 surface sediment samples (SS), by the 16S rRNA gene sequencing. The mean value of clean reads for each sample was 65,149 ± 6930. The rarefaction curves based on the OTUs (Figure S1) showed that all the gut content samples and five of the sediment samples tended to approach the saturation plateau. The Good's coverage of all the samples was over 97.80% (Table 1), indicating that the sequencing depth was reliable and could truly reflect the composition of most of the bacteria in the samples.

**Table 1.** Number of OTUs and Good's coverage for different samples.

| Group * | Sample ID | OTU | Good's Coverage | Group * | Sample ID | OTU | Good's Coverage |
|---------|-----------|-----|-----------------|---------|-----------|-----|-----------------|
| SM | SM1 | 1551 | 99.30% | | SC1 | 1659 | 98.70% |
| | SM2 | 1502 | 99.60% | | SC2 | 1909 | 98.80% |
| | SM3 | 1481 | 99.30% | SC | SC3 | 2351 | 98.80% |
| | SM4 | 1869 | 99.20% | | SC4 | 1782 | 99.00% |
| | SM5 | 1739 | 99.00% | | SC5 | 1858 | 98.90% |
| | HA1 | 1792 | 98.90% | | SS1 | 4224 | 98.00% |
| | HA2 | 2923 | 98.30% | | SS2 | 4272 | 98.30% |
| HA | HA3 | 2942 | 98.40% | SS | SS3 | 4086 | 98.20% |
| | HA4 | 2621 | 98.70% | | SS4 | 4210 | 97.80% |
| | HA5 | 2169 | 98.70% | | SS5 | 4017 | 97.90% |

\*: SM, SC, and HA are sample groups that represent the gut contents of *S. monotuberculatus*, *S. chloronotus*, and *H. atra*, respectively, and SS represents the surrounding sediments.

A total of 10,523 OTUs were obtained from the 20 samples (Table 1). The Venn diagram based on the results of the OTU cluster analysis indicated that 1999 OTUs (19.12%) were shared by all the gut contents and surrounding sediment samples and 2172 OTUs (28.38%) were shared by the gut contents of the 3 species of sea cucumbers (Figure 1).

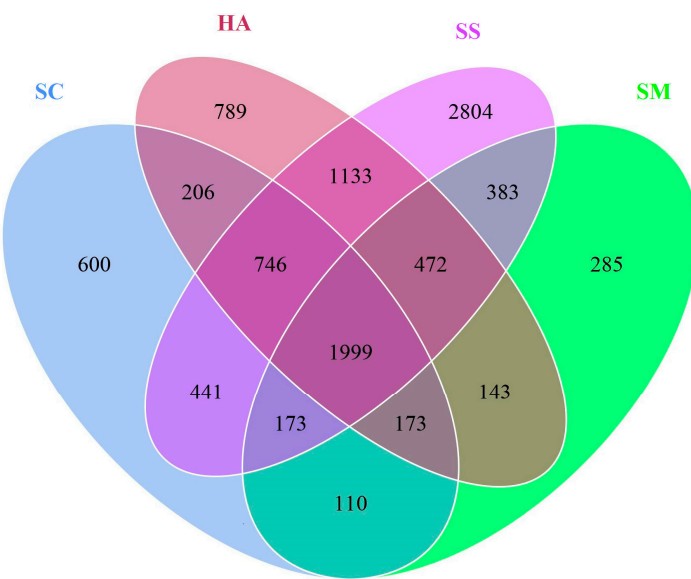

**Figure 1.** OTUs Venn diagram about microbial communities in sediment and the gut.

The diversity and richness indices of sediment and gut microbiota are shown in Figure 2. The ACE index and the Chao1 index were used to quantify the richness. The diversity of samples was calculated via the Shannon index and the Simpson index. The two indices of each part were double-checked to make our results more reliable. The method has been successfully performed in community diversity research [42,43]. The richness and

diversity of microbiota in sediments were significantly higher than those in gut microbiota ($p < 0.01$). In the gut samples, the richness of microbial communities in the SM group was lower than that in the HA group ($p < 0.05$). In addition, there was no significant difference in gut microbiome diversity between the three sea cucumber species.

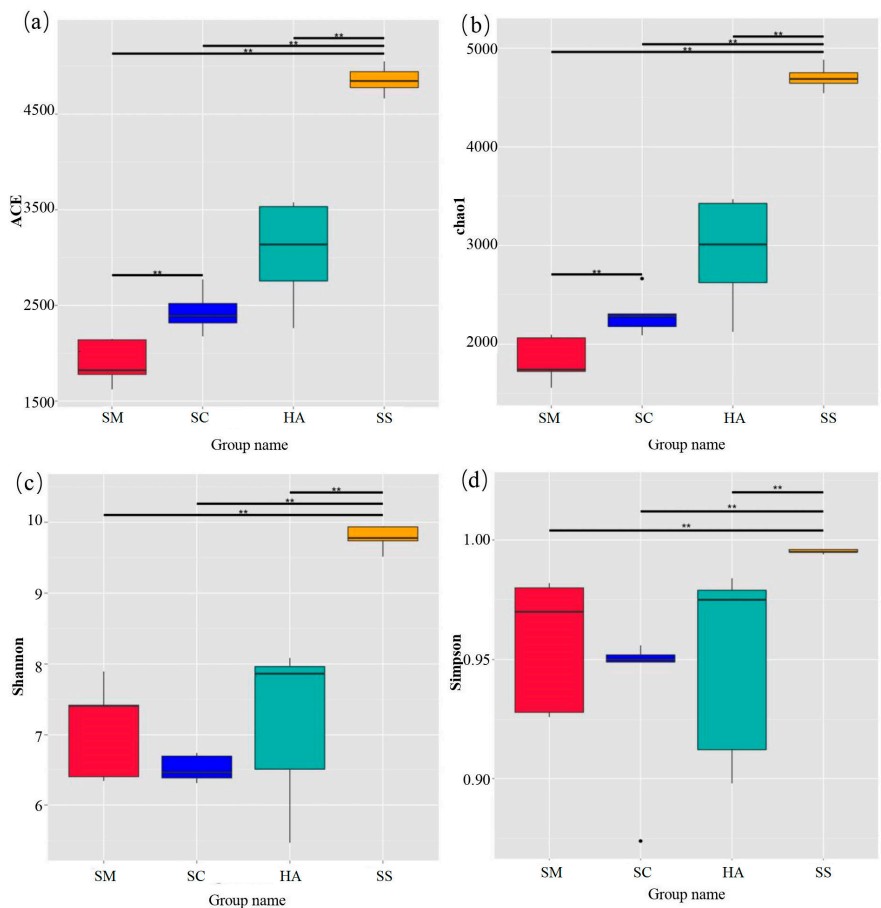

**Figure 2.** Alpha diversity index of microbiota in sediments and the gut. (**a**) ACE estimator, (**b**) Chao1 estimator, (**c**) Shannon index, and (**d**) Simpson index. ** $p < 0.01$.

### 3.2. Relationships of Microbial Communities among the Gut and Sediment Samples

Beta diversity analysis can reflect the similarities and differences in the microbial communities between samples [44,45]. At the OTU level, two-dimensional principal coordinate analysis (PCoA) was performed for all samples based on the weighted Unifrac distance, with PC1 accounting for 50.03% of the total variation and PC2 for 11.97% (Figure 3a). As shown in the figure, the four groups of samples were clustered separately into two groups: all the sediment samples, and all the gut content samples from the three sea cucumber species. Anosim analysis is a nonparametric test used to test whether the differences between groups are significantly greater than the differences within groups, to determine whether the groups are meaningful [46]. The Anosim analysis showed that the intra-group differences were less than the inter-group differences (R > 0) (Table 2). There were significant differences between the groups ($p < 0.05$), except that there was no significant difference between the HA and SC groups.

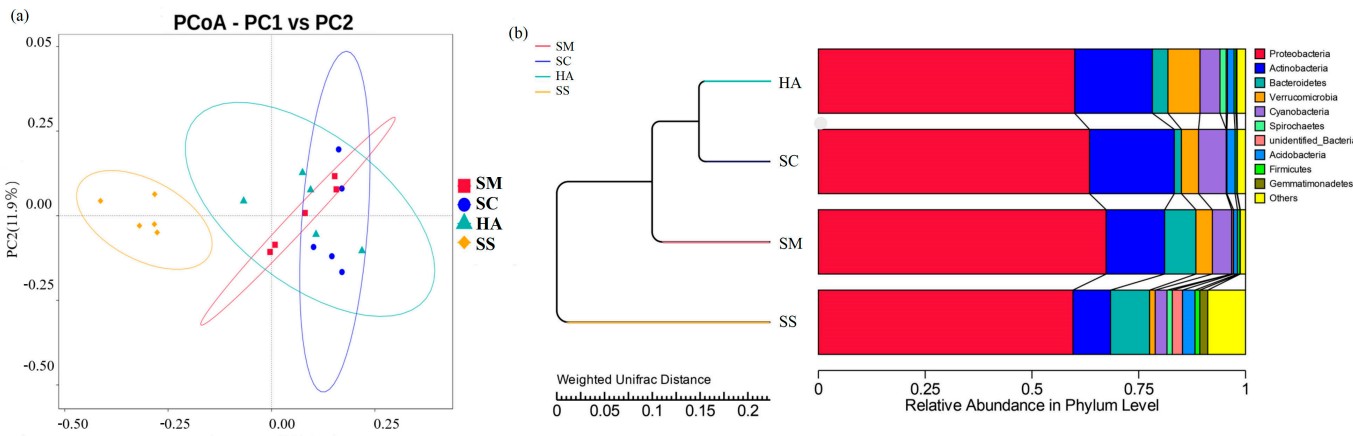

**Figure 3.** Beta diversity analyses of the bacterial communities of sediments and the guts of sea cucumber. (**a**) 2D principal coordinates analysis of the bacterial communities of different samples based on the weighted Unifrac distance at the OTU levels. (**b**) UPGMA tree showing the similarities of the bacterial community structures among the different samples.

**Table 2.** Anosim analyses for different samples.

| Group | R-Value | *p*-Value | Group | R-Value | *p*-Value |
| --- | --- | --- | --- | --- | --- |
| SM-SS | 1.000 | 0.008 | SC-SM | 0.500 | 0.009 |
| SC-SS | 1.000 | 0.010 | HA-SM | 0.628 | 0.009 |
| HA-SS | 0.948 | 0.015 | HA-SC | 0.020 | 0.310 |

The UPGMA clustering tree at the phylum level based on the weighted Unifrac distance displayed the similarity of the microbial communities among the different samples (Figure 3b). The analysis showed that the samples were firstly divided into two groups: one group consisted of all sea cucumber (SM, SC, HA) samples, and the other group was composed of all the environmental sediment samples (SS). The first group was further divided into three clusters, corresponding to the three species of sea cucumbers' gut microbial community grouping, indicating that the gut microflora community of different sea cucumbers had different characteristics.

### 3.3. Relative Abundance of Microbial Communities

An average of 24.6 ± 2.74, 30.0 ± 2.08, and 34.8 ± 2.41 phyla were identified from the gut microbiota of *S. monotuberculatus*, *S. chloronotus*, and *H. atra*, respectively, and there were 54 phyla in the sediment microbial samples. The relative abundance of the 10 most abundant phyla in the sediment and the gut of the three sea cucumber species are shown in Figure 4. The dominant phyla in the *S. monotuberculatus* gut microbiota were Proteobacteria, Actinobacteria, and Bacteroidetes, with relative abundances of 67.37 ± 7.04%, 13.70 ± 1.47%, and 7.32 ± 6.10%, respectively, and the dominant phyla of gut microbes in *S. chloronotus* were Proteobacteria, Actinobacteria, and Cyanobacteria, with relative abundances of 63.48 ± 10.90%, 19.85 ± 14.04%, and 6.48 ± 3.28%, respectively. The dominant phyla of the *H. atra* gut microbes were Proteobacteria, Actinobacteria, and Verrucomicrobia, with relative abundances of 60.00 ± 5.75%, 18.19 ± 6.42%, and 7.50% ± 0.60%, respectively. The dominant microbial phyla of the sediment group were Proteobacteria, Bacteroidetes, and Actinobacteria, with relative abundances of 59.60 ± 5.22%, 9.15 ± 6.42%, and 8.79 ± 3.36%, respectively. Proteobacteria was the predominant phylum in all samples, and there was no significant difference in the relative abundance among all groups. Actinobacteria was also a dominant phylum of all samples, with the relative abundance of SM sample groups being lower than that of the other three groups ($p < 0.05$, q < 0.05).

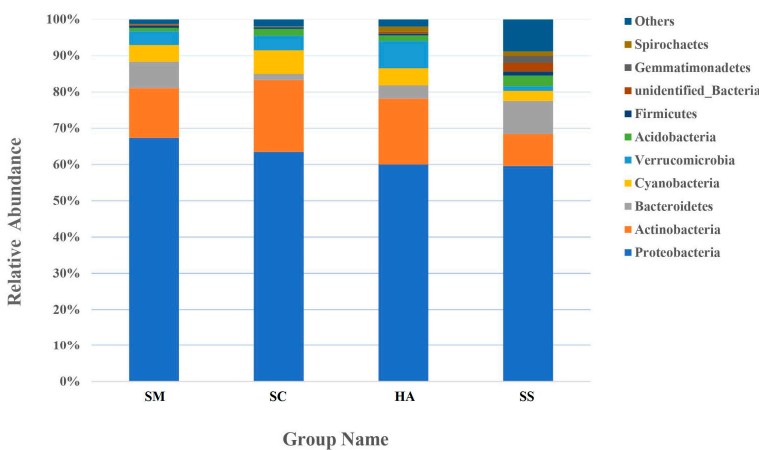

**Figure 4.** The relative abundance of the 10 most abundant phyla.

The dominant families in the gut microbiota of SM, SC, and HA were sequences related to Rhodobacteraceae (31.76 ± 13.77%) and Halieaceae (8.66 ± 3.23%), Rhodobacteraceae (23.49 ± 6.16%) and Burkholderiaceae (15.91 ± 11.15%), and Rhodobacteraceae (19.67 ± 5.68%) and Burkholderiaceae (12.03 ± 11.82%), respectively (Figure 5). The top-two dominant families in sediment samples were Desulfobacteraceae (8.25 ± 2.63%) and Desulfobulbacea (7.46 ± 2.36%). Rhodobacteraceae was the most abundant family in all the gut content samples and was higher than the relative abundance in the sediment (3.88 ± 1.52%, $p < 0.05$, q < 0.05). The relative abundance of Desulfobulbaceae and Desulfobacteraceae in SS was significantly higher than that in SM, SC, and HA ($p < 0.01$, q < 0.05).

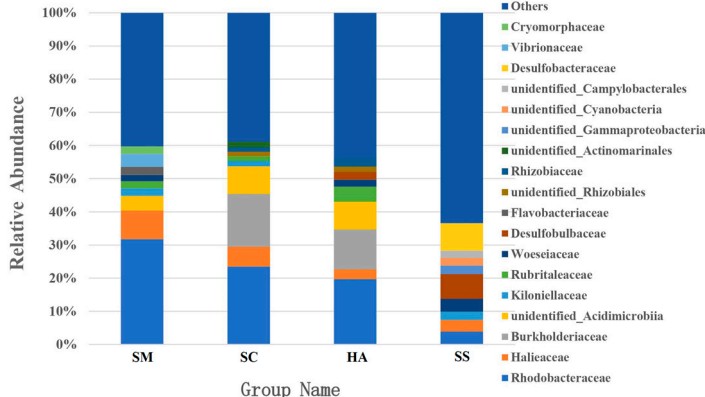

**Figure 5.** The relative abundance of the 10 most abundant families in each group.

A large proportion (47.48–72.96%) of the reads in all the libraries could not be classified at the genus level. The ten most abundant genera in the different samples accounted for 59.70%, 61.11%, 56.11%, and 36.58% of reads in the SM, SC, HA, and SS libraries, respectively (Figure 6). *Ruegeria* was the abundant genus in all gut samples, SM (16.52 ± 7.01%), SC (12.09 ± 3.47%), and HA (10.12 ± 4.45%), respectively, higher than in sediments (1.57 ± 0.71%, $p < 0.05$, q < 0.05). The genus with the most relative abundance of SC (15.73 ± 11.09%) and HA (11.91 ± 11.80%) was *Ralstonia*, while the abundance was low in the SM (1.42 ± 1.54%) and SS (1.18 ± 1.46%) groups. *Woeseia* was the most abundant genus in sediments (3.72 ± 0.84%), significantly higher than in gut samples (SM, 1.80 ± 0.74%; SC, 1.03 ± 0.36% ($p < 0.01$, q < 0.05)).

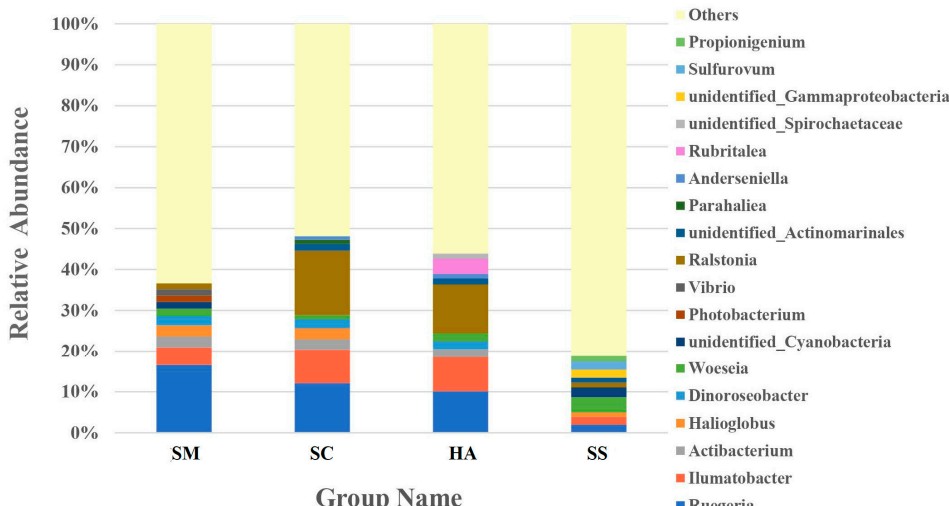

**Figure 6.** The relative abundance of the 10 most abundant genera in each group.

Table S1 lists the genera whose abundance was greater than 0.005% in all groups. *Enterovibrio* was the genus with the highest abundance in the SM (0.73%) group in Table S1, while the abundance was very low in the rest. Some notable genera were also found among these low-abundance genera, such as *Lactobacillus* (SM, 0.0065%; SC, 0.0249%; HA, 0.0584%; SS, 0.000135%) and *Pseudomonas* (SM, 0.0086%; SC, 0.0049%; SS, 0.0037%), whose relative abundance was not high, but was higher in the gut than in the sediment.

Based on the *t*-test, it was possible to analyze the differences in the gut microbiota between the three species of sea cucumber. At the phylum level, the number of phyla with differential gut microflora of the three sea cucumber species was low and not statistically significant (Figure 7a). Specifically, the only significant differences in the phylum were Acidobacteria ($p < 0.05$, q < 0.05) between SM and SC groups, and Schekmanbacteria ($p < 0.05$, q < 0.05) between SC and HA groups. The main differences in gut microbiota among the three species were seen at the family level (Figure 7b) and the genus level (Figure 7c). At the family level, there were five families with differences between SM and HA ($p < 0.05$, q < 0.05). At the genus level, five genera were different between SM and HA ($p < 0.05$, q < 0.05).

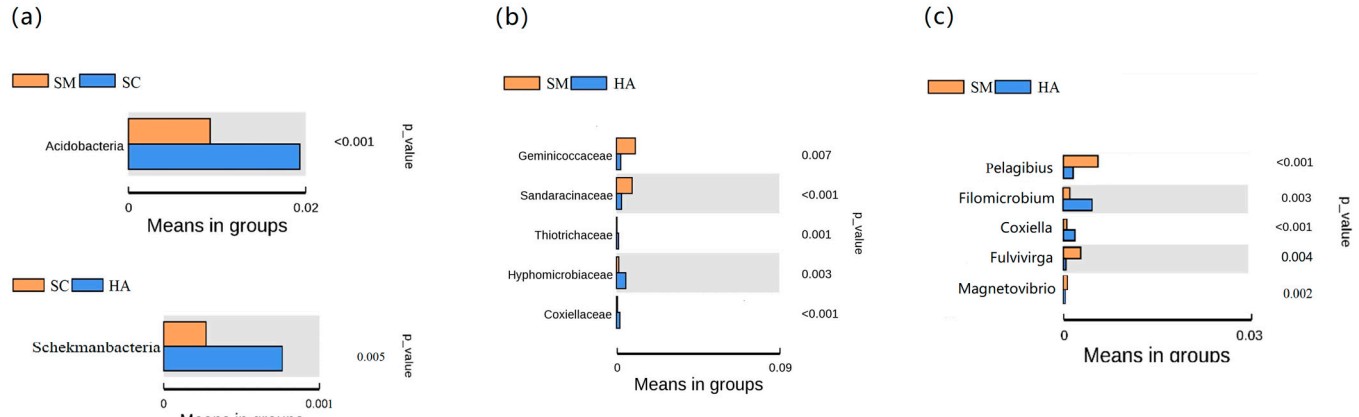

**Figure 7.** Identified differentially abundant taxa between samples via the *t*-test ($p < 0.05$, q < 0.05). (**a**) At the phylum level, (**b**) at the family level, and (**c**) at the genus level. The left of the figures shows the differential abundance between groups, with each bar representing the mean value. On the right are the *p*-values for the between-group significance tests for the corresponding differential species.

### 3.4. Annotated Analysis of Microbiota Function

Based on the KEGG pathway database, a total of 328 level-3 pathways were annotated in the 4 groups of samples. The *t*-test showed that there were significant differences in the level-3 pathways of gut microbiota and sediment microbiota among the 3 sea cucumber species, showing 178, 213, and 132 highly significantly different ($p < 0.001$) pathways between SS and SM, SS and SC, and SS and HA, respectively. There were 12, 15, and 8 significantly different pathways between SM and SC, SM and HA, and SC and HA, respectively ($p < 0.01$). Cluster analysis was performed for the top 34 level-3 metabolic pathways in relative abundance (Figure 8). The four groups of samples were divided into two major groups, in which the SS group was a group, and the gut sample groups were clustered into a group, indicating that the functional similarity between the gut microflora of the three sea cucumber species was found to be closer than that of the sediment microflora. The main pathways in the gut include transporters, ABC transporters, transcription factors, arginine and proline metabolism, butanoate metabolism, propanoate metabolism, pyruvate metabolism, valine, leucine, and isoleucine degradation, glycine, serine, and threonine metabolism, and fatty acid metabolism (Figure 8). The PCA revealed that the function of gut microflora in sea cucumbers was markedly different from that of sediment microflora. In addition, the functions of the gut microbiota in the three species of sea cucumber were clustered together and did not completely overlap with each other (Figure 9).

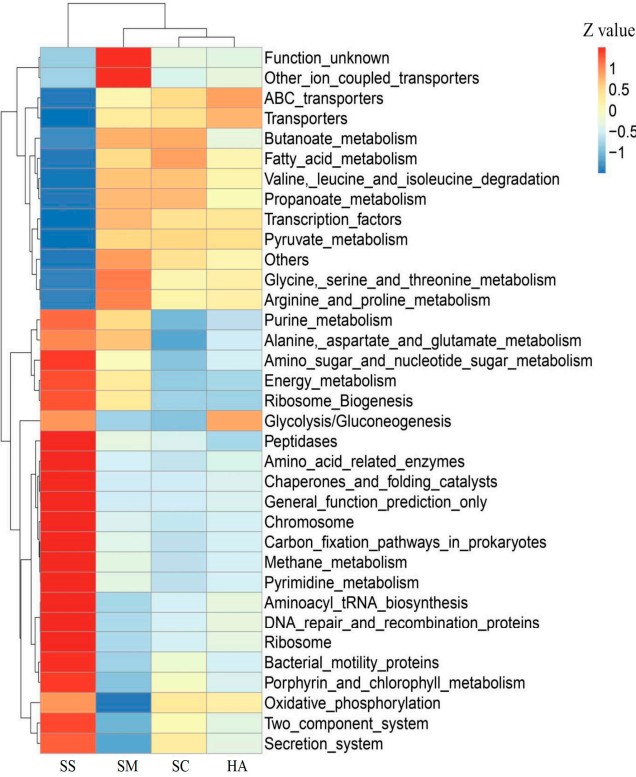

**Figure 8.** Level-3 cluster heatmap based on the Tax4Fun annotation of the functional relative abundance of different samples. The z-value is the difference between the relative abundance of one sample and the mean relative abundance of all samples at that classification, divided by the standard deviation of all samples at that classification.

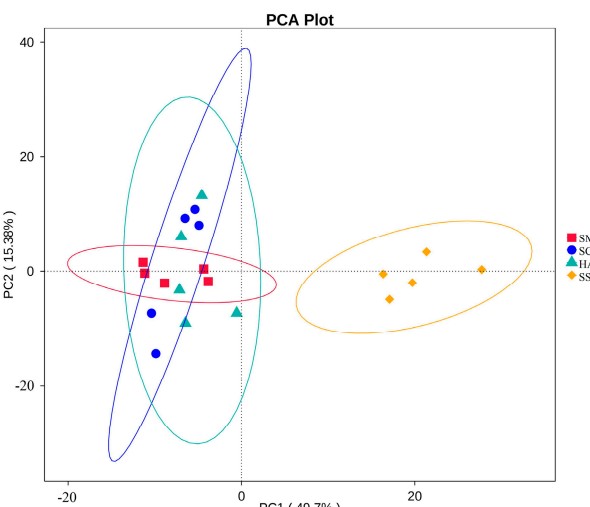

**Figure 9.** PCA maps of KEGG-annotated level-3 functional abundance for sediments (SS) and sea cucumber gut.

## 4. Discussion

### 4.1. Dominant Bacteria in the Gut Microbiota

In this study, Proteobacteria, Actinobacteria, and Bacteroidetes were the most abundant phyla in the gut microbiota of the three species of sea cucumbers and the ambient sediments. Gao et al. [47] also found that Proteobacteria and Bacteroidetes are the dominant bacteria in the intestinal tract of *H. atra*, which was consistent with the present study. Zhang et al. [18] isolated various aerobic bacteria from the gut of *H. leucospilota* for molecular identification and found that most of the isolates belonged to Firmicutes and Proteobacteria. At the family level, Rhodobacteraceae was the most abundant microbial community in the gut of the three sea cucumber species, higher than the sediment microflora. Pujalte et al. [48] revealed that the family Rhodobacteraceae was a significant microflora coexisting with the host and is highly engaged in carbon/sulfur cycles in the gut. The ability of most Rhodobacteraceae members to produce vitamin B12 has also been confirmed, which is needed for the growth of invertebrates [49]. We further explored the genus level and found that *Ruegeria* was the most abundant genus in the gut contents of *S. monotuberculatus*, which was the second most abundant genus in the guts of *S. chloronotus* and *H. atra*. *Ruegeria* belongs to the phylum Proteobacteria in the family Rhodobacteraceae. Some researchers revealed that *Ruegeria* is one of the most common genera in marine environments, and is generally isolated from water, sediments, and marine organisms in marine environments [50–54], which may account for its high abundance in the guts of sea cucumbers.

### 4.2. Potential Probiotics

Probiotics are defined as live microorganisms that positively affect host animals by maintaining the microbiological balance in the gut [55–57]. Probiotics used in aquaculture can be divided into two categories according to their mechanism of action. The most common group of probiotics can be added to aquatic animal feed [58–61], and the others can be used as ecological regulators to improve water quality [62,63]. In this study, *Lactobacillus iners* and *Lactobacillus reuteri* were found to exist in the guts of the three sea cucumber species. It is well-established that *Lactobacillus* has high colonization properties and thus remains for a longer time on the fish gut epithelial surface, conferring great beneficial effects to the host and the gut microbiota [64]. *L. reuteri* has numerous benefits for hosts, including promoting host health, reducing infections, improving feed tolerance, increasing the absorption of nutrients, minerals, and vitamins, modulating host immune responses, and promoting gut mucosal integrity [65–69]. In addition to *Lactobacillus*, the *P. geniculata* and the *P. stutzeri* of the genus *Pseudomonas* were present in this study. The genus *Pseudomonas* includes Gram-negative aerobic bacteria or facultative symbiotic anaerobes. The genus *Pseudomonas*

was reported to be a potential probiotic, present in the gastrointestinal tract of fish and shellfish [56,70]. Furthermore, *Pseudomonas* was very effective against the 'infectious hematopoietic necrosis virus' in the fish gut [71]. In recent years, probiotics that were more commonly applied in aquaculture mainly belonged to the genera *Lactobacillus*, *Bacillus*, and *Pseudomonas* [62,72–76]. At present, one of the problems is how to deal with the diseases of sea cucumber culture in a healthy, safe, legal, and effective way. However, there are relatively few practical studies on the application of microecological preparations in sea cucumber culture at present, and there is a lack of practice and innovation. In this study, *Lactobacillus* and *Pseudomona* were detected in the guts of the three sea cucumber species, suggesting that they could be used as candidate probiotics. Future studies should evaluate the abundance and function of putative probiotics in the sea cucumber gut.

### 4.3. Relationship between Microbial Communities in the Gut Contents and the Sediments

In this study, we compared the composition of microbial communities between the gut contents of the three sea cucumber species and the surrounding sediments. The microflora richness and diversity in the guts of all three sea cucumber species were significantly lower than that in the sediments ($p < 0.01$). The PCoA score plot and heatmap analysis also showed that the gut samples of sea cucumbers and their surrounding sediment samples were clustered in different groups. We speculated that there may be two reasons for the large differences in the composition of the gut communities and the environmental sediment microflora. The first is that the gut environment is only suitable for the reproduction of certain microbes. In addition, studies have shown that the gut is a relatively hypoxic environment [77,78]. Zhang et al. [18] found that the isolated aerobic isolates from the gut of *H. leucospilota* were potentially tolerant to anaerobic conditions in the intestine of holothurians. In this study, the concentrations of *Woeseia* and *Sulfurovum* in the sediment samples were significantly higher than those in the gut samples of the three sea cucumber species ($p < 0.05$). Most of the *Woeseia* and *Sulfurovum* are chemoautotrophic bacteria, which require sulfur or light as energy sources [79,80]. There was a lack of light and sulfur in the guts in this study, and therefore, we speculated that the special gut environment may lead to the significantly lower relative abundance of these two bacterial groups in the guts. Furthermore, they, as part of the sea cucumbers' food contents, might be digested in the digestive tract.

Secondly, there are some specific bacteria enriching or colonizing the guts of sea cucumbers. The gut of *A. japonicus* contains a large number of microbial communities that could produce a variety of enzymes, with the function of digestion and transformation of nutrients [81]. In this study, the bacteria of genera *Ruegeria* and *Actibacterium* were found to be enriched in the guts of the three species of sea cucumber (Figure 6). Porsby et al. [82] found that *Ruegeria* can antagonize many pathogenic bacteria, such as *Vibrio anguillarum*, by producing the antibiotic tropodithietic acid. Studies have shown that *Ruegeria* can be used as a potential macromolecule polysaccharide degradation bacterium [83,84]. We speculate that the presence of these microorganisms contributes to the digestion of the sea cucumber and improves its disease resistance. In this study, based on KEGG level-3 pathway analysis, we found that pathways related to metabolic functions, including arginine and proline metabolism, butanoate metabolism, propanoate metabolism, pyruvate metabolism, glycine, serine, and threonine metabolism, and fatty acid metabolism, were significantly upregulated in the guts of the three sea cucumber species, compared to the sediment microflora. It also implied that the gut microbiota was actively involved in the daily metabolic process of the host. This led to the enrichment of certain microflora in the gut of sea cucumbers, and their abundance was higher than that in the sediments.

### 4.4. Relationships among Gut Microbial Communities of Three Sea Cucumber Species

We also compared the gut microflora among the three sea cucumber species. The PCoA score map showed that the gut microbiota of the three sea cucumber species were clustered together and there was no apparent grouping. At the phylum level, the top-four dominant

microbiota of the three sea cucumber species were Proteobacteria, Actinobacteria, Bacteroidetes, and Verrucomicrobia. Mfilinge et al. [85] found that the fatty acid compositions in the guts of *H. atra* and *H. leucospilota* were similar, and they considered that ingestion of the same type of organic material was the primary reason. In this study, we speculated that the high similarity in the bacterial community composition in the gut microbiota between the three sea cucumber species was probably due to the same maritime space. Although the same environment led to similar microbial compositions at the phylum level, the Anoism analysis, *t*-test, and Alpha diversity results showed that there were differences in the gut microbial communities among the three sea cucumber species, mainly at the family and genus levels. It was speculated that the habitat preference and feeding choice of the three sea cucumber species make their gut microbiota different. *S. monotuberculatus* mainly live near coral reefs or under rocks, while the other two species mainly live on the sandy bottom, where the water flows gently [3,86]. The structural relationship of gut microflora in the *H. atra* and *S. chloronotus* was much more similar than that with *S. monotuberculatus* in our study (Figure 3b). Presumably, the sea cucumbers' habitats have an effect on their gut microorganisms.

Some reports have claimed that sea cucumbers feed selectively, particularly concerning particle size [12,87,88]. The structure of their tentacles may affect the feeding of sea cucumbers [12], and the diameter of the mastoid cluster of *S. monotuberculatus* is smaller than that of the *H. atra*, which is suitable for the uptake of fine sediment particles [31,89]. Previous studies showed that *H. atra* likes to feed on gravel and coarse sand [30], but *S. chloronotus* likes to feed on fine sand [3,22]. Moriarty [13] commented that *H. atra* and *S. chloronotus* chose sediment components with a high bacterial content. Some studies indicated that *H. atra* fed on sediments with less microalgal biomass compared to *S. chloronotus*, and the latter species also selected sediment patches with finer particles than the former, which showed the different feeding strategies between the two species [90,91]. Sea cucumbers also show their feeding preferences in other ways, such as the bacterial biomass, community composition, and organic matter content of sediments [13,92,93]. In future studies, well-controlled experiments must be conducted to test this speculation.

## 5. Conclusions

Here, we used a high-throughput 16S rRNA gene-based molecular microbiology approach to study the gut bacterial communities of three species of tropical sea cucumbers and sediment microflora. Lactobacillus and Pseudomonas were detected in the gut of the three sea cucumber species studied, suggesting that they could be used as candidate probiotics. Future studies should evaluate the abundance and function of the potential probiotics in sea cucumbers. Based on the KEGG metabolic pathway analysis, the gut microbiome of sea cucumbers plays a key role in gut metabolism, digestion, and absorption. We also observed distinct differences in the bacterial communities of the gut and sediment microflora of the three sea cucumber species studied. The unique gut environments likely contribute to these differences. Additionally, the gut microbiome also showed differences due to the different habitats of the three tropical sea cucumber species. It has been speculated that selective feeding may be the main reason for the different microbial communities in the gut microbiome of different sea cucumbers. This study shed a microscopic light on the differences in the diets of different sea cucumbers and the relationship between gut and habitat microbes. These results provide important data for future research on the gut microbiota of tropical sea cucumbers.

**Supplementary Materials:** The following supporting information can be downloaded at: https://www.mdpi.com/article/10.3390/d15070855/s1, Figure S1: Rarefaction curve of the gut content samples and sediment samples *: SM, SC and HA are sample groups that represent the gut contents of *S. monotuberculatus*, *S. chloronotus* and *H. atra*, respectively, and SS represents the surrounding sediments. Table S1: Genus level relative abundance greater than 0.5% in each sample.

**Author Contributions:** Conceptualization, F.G.; methodology, F.G.; software, Y.W. (Yanan Wang); validation, C.J. and Y.W. (Yanan Wang); formal analysis, C.J.; investigation, Y.W. (Yanan Wang), F.G. and Y.Z.; resources, F.G.; data curation, Y.W. (Yanan Wang) and C.J.; writing—original draft preparation, Y.W. (Yanan Wang); writing—review and editing, Y.W. (Yanan Wang), F.G. and C.J.; visualization, Y.W. (Yanan Wang), Y.R., Y.W. (Yuanhang Wang) and Z.X.; supervision, F.G.; project administration, F.G.; funding acquisition, F.G. and Q.X. All authors have read and agreed to the published version of the manuscript.

**Funding:** This research was funded by the National Natural Science Foundation of China (Nos. 42166005, 42076097), the Hainan Provincial Key Research and Development Program (ZDYF2021XDNY130), the Natural Science Foundation of Hainan Province (321RC1023), and the State Key Laboratory of Marine Resource Utilization in South China Sea Open Project (MRUKF2021008).

**Institutional Review Board Statement:** Not applicable.

**Data Availability Statement:** The datasets presented in this study can be found in online repositories at: https://www.ncbi.nlm.nih.gov/ (accessed on 8 November 2022), PRJNA898248.

**Acknowledgments:** We sincerely thank the reviewers for their critiques and suggestions.

**Conflicts of Interest:** The authors declare that the research was conducted in the absence of any commercial or financial relationship.

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
