# Peer review of "Comparative Analysis of Gut Microbial Community Structure of Three Tropical Sea Cucumber Species"

_diversity, doi:10.3390/d15070855_

Round 1
Reviewer 1 Report
This study characterized the gut microbiota of three species of sea cucumber in comparison to each other and to surface sediment from the environment in which they were collected. The richness and diversity of the microflora were lower in the sea cucumber guts than in the sediment, and there were some differences among the three species in microflora composition. The authors speculate that the difference in microbiota between sea cucumber guts and sediment may be explained by the different environment provided by the gut, whereas the difference among species may have to do with selective feeding. Apparently Stichopus chloronotus and Holothuria atra were collected from the sediment surface, whereas Stichopus monoturberculatus were found under rocks in the cracks and crevices of reef. The gut microflora of the first two species was much more similar than the microflora of S. monoturberculatus. Some sea cucumber species are deposit feeders and feed directly on surface sediment, and others are filter feeders, collecting material from the water column. Given where these species were collected, I am wondering if differences in feeding strategy could explain the differences in gut microflora. My guess is that S. chloronotus and H. atra may be deposit feeders whereas S. monoturberculatus may be a filter feeder? If there is evidence of this, it should be discussed.
Some specific comments follow:
L 105-106: Why did the authors choose the abbreviations HC, LC, and SC for the three species, which are impossible to remember? It would have been more logical to choose SM, SC, and HA. Likewise, why is the surface sediment abbreviated DN, rather than SS?
L 112: It is stated that the surface sediment was collected with a syringe. To what depth and what diameter was the opening of the syringe? Were any other analyses of the sediment done, such as organic matter or carbon content, grain size distribution, etc?
L 116: Presumably the gut samples were taken from the same portion of the gut for all of the replicates. This could be important if conditions along the length of the gut vary, which is possible. Please confirm.
L 158-159: Please define alpha and beta diversity at first use.
L 160-171: Why did you use so many different metrices for the same thing? Did they tell you different things about alpha diversity? Are there advantages and disadvantages to each? These issues should be discussed in the Discussion.
L 197-199: It appears that this text refers to instructions for writing the section and should be removed.
L 201-203: “The clean reads numbers for each sample ranged from 54, 498 to 77, 831 with a mean average of 65, 149 +/- 6930.” This sentence is impossible to understand. To what exactly do all of these numbers refer? Please rewrite for clarity.
L 229: OUT should be OTU.
L 258: were should be are.
Figure 4 caption: phylum should be phyla.
Figure 5 caption: there are more than 10 families listed.
L 280: “of microbial” should be deleted.
Figure 6 caption: There are more than 10 genera listed.
L 308-309: One of these two comparisons should be HC and SC. Please correct.
Figure 7: Were all of these t-tests corrected for experiment-wise error so as not to inflate the Type I error rate? Simply by making so many tests, some will come up as significant by chance. Therefore the significance level (0.05) is often divided by the number of comparisons to avoid this problem. So for 10 comparisons, only those having a P-value less than 0.005 would be considered significant.
L 341: Should read “and did not completely overlap with each other”.
L 407: Whether oxygen is low in the sediment will depend on the depth at which it is sampled (which is why I asked above). In organic-rich sediments, the depth of the oxidized zone may only be a few mm.
L 459-460: This sentence is repeated twice. Please delete.
L 483-485: Consider reworking the last sentence of the Conclusions for a more significant take-home message.
The English is pretty good, but could use some polishing by the editorial office.
Author Response
Response to Reviewer 1 Comments
This study characterized the gut microbiota of three species of sea cucumber in comparison to each other and to surface sediment from the environment in which they were collected. The richness and diversity of the microflora were lower in the sea cucumber guts than in the sediment, and there were some differences among the three species in microflora composition. The authors speculate that the difference in microbiota between sea cucumber guts and sediment may be explained by the different environment provided by the gut, whereas the difference among species may have to do with selective feeding. Apparently Stichopus chloronotus and Holothuria atra were collected from the sediment surface, whereas Stichopus monoturberculatus were found under rocks in the cracks and crevices of reef. The gut microflora of the first two species was much more similar than the microflora of S. monoturberculatus. Some sea cucumber species are deposit feeders and feed directly on surface sediment, and others are filter feeders, collecting material from the water column. Given where these species were collected, I am wondering if differences in feeding strategy could explain the differences in gut microflora. My guess is that S. chloronotus and H. atra may be deposit feeders whereas S. monoturberculatus may be a filter feeder? If there is evidence of this, it should be discussed.
Reply: By reviewing the literature and combining the results of previous studies on the feeding of these three species of sea cucumber, we conclude that they are all deposit-feeding sea cucumbers. Described the sediment preferences of these three species in previous studies in the discussion. Please refer to lines 481-491 for details.
References:
Liao, Y. Fauna Sinica: Echinoderma: Holothuroidea, 1st ed.; Science Press: Beijing, 1997; pp. 99-156. (In Chinese)
Roberts, D. Deposit-feeding mechanisms and resource partitioning in tropical holothurians. J. Exp. Mar. Biol. Ecol. 1979, 37, 43-56. Doi:10.1016/0022-0981(79)90025-X
Purcell, S.W.; Samyn, Y.; Conand, C. Commercially important sea cucumbers of the world. 1st ed.; FAO: Rome, 2012: pp. 42-104.
Dissanayake, D.; Stefansson, G. Habitat preference of sea cucumbers: Holothuria atra and Holothuria edulis in the coastal waters of sri lanka. J. Mar. Biol. Assoc. UK. 2012, 92, 581-590. Doi:10.1017/S0025315411000051
Xue, Y.; Gao, F.; Xu, Q.; Huang, D.; Wang, A.; Sun, T. Study on feeding selection of environmental sediment and digestive adaptability of Holothuria atra. Oceanologia et. Limnologia Sinica 2019, 50, 1070-1079. Doi:10.11693/hyhz20190200033. (In Chinese).
Some specific comments follow:
Point 1: L 105-106: Why did the authors choose the abbreviations HC, LC, and SC for the three species, which are impossible to remember? It would have been more logical to choose SM, SC, and HA. Likewise, why is the surface sediment abbreviated DN, rather than SS?
Reply: Thanks for the advice of the experts, the previous naming method is in accordance with our language habit, which is really a lack of professionalism. The full text has been revised: HC, LC, and SC are changed to SM, SC, and HA. Change DN to SS.
Point 2: L 112: It is stated that the surface sediment was collected with a syringe. To what depth and what diameter was the opening of the syringe? Were any other analyses of the sediment done, such as organic matter or carbon content, grain size distribution, etc?
Reply: Thanks to the experts' suggestions. ​The acquisition depth of the syringe at the time of sampling was less than 1 cm and the diameter of the opening was 2.9 cm, and the corresponding instructions were added at the corresponding location in the article. However, for other analyses of the sediment, due to the purpose of this experiment and the analysis of the bacterial community, the relevant data are really missing in this experiment. In the following experiment, we will take your opinions into account to further analyze other components.
L 110-113: As a result, the surface sediments (SS, n=5) we collected were located at the confluence of the two habitats were collected using the 50-ml syringe (the diameter of the adjusted syringe opening was 2.9 cm, and the collection depth was less than 1cm).
Point 3: L 116: Presumably the gut samples were taken from the same portion of the gut for all of the replicates. This could be important if conditions along the length of the gut vary, which is possible. Please confirm.
Reply: Thanks for the advice of the experts, we took the intestinal contents within 2-3cm of the foregut of the sea cucumber, and added the corresponding description in the corresponding position in the article.
L 118-120: The gut contents were squeezed out from the digestive tract (foregut 2-3cm) and collected in a sterile cryotube.
Point 4: L 158-159: Please define alpha and beta diversity at first use.
Reply: We are grateful to the reviewers for their suggestions. We add the corresponding explanation in the corresponding place.
L 159-162: The subsequent alpha diversity analysis (to analyze the diversity of microbial communities within the sample) and beta diversity analysis (to compare the microbial community structure of different samples) was based on data after normalization.
Point 5: L 160-171: Why did you use so many different metrices for the same thing? Did they tell you different things about alpha diversity? Are there advantages and disadvantages to each? These issues should be discussed in the Discussion.
Reply: Thank you for your suggestion. There is a corresponding explanation in lines 227-231 of the original text, and we have also added an explanation. The 4 diversity indices (Shannon, Simpson, Chao1, Ace) in Figure 2 are part of the Alpha diversity analysis. Indices Shannon and Simpson were applied to evaluate species diversity, and indices Chao1 and Ace were applied to evaluate species richness. The two indices of each part were double check to make our results more convincing. The method has been successfully performed in community diversity research (Forbes et al., 2016; Fan et al., 2019).
References:
Forbes, J. D., Van Domselaar, G., & Bernstein, C. N. (2016). Microbiome survey of the inflamed and noninflamed gut at different compartments within the gastrointestinal tract of inflammatory bowel disease patients. Inflammatory bowel diseases, 22(4), 817-825.
Fan, L., Wang, Z., Chen, M., Qu, Y., Li, J., Zhou, A., ... & Zou, J. (2019). Microbiota comparison of Pacific white shrimp intestine and sediment at fresh water and marine cultured environment. Science of the Total Environment, 657, 1194-1204.
Point 6: L 197-199: It appears that this text refers to instructions for writing the section and should be removed.
Reply: Thanks for the reviewer’s useful comments. We have corrected the mistake in the revised manuscript, and it is an important reminder to us that we will examine our manuscripts more carefully in the future.
Point 7: L 201-203: “The clean reads numbers for each sample ranged from 54, 498 to 77, 831 with a mean average of 65, 149 +/- 6930.” This sentence is impossible to understand. To what exactly do all of these numbers refer? Please rewrite for clarity.
Reply: Thank you for your precious comments and advice. We have revised the manuscript accordingly.
L 212: The mean value of clean reads for each sample was 65,149 ± 6930.
Point 8: L 229: OUT should be OTU.
Reply: Thanks for the reviewer’s useful comments. We have corrected the mistake in the revised manuscript, and it is an important reminder to us that we will examine our manuscripts more carefully in the future.
Point 9: L 258: were should be are.
Reply: Thanks for the reviewer’s useful comments. We have corrected the mistake in the revised manuscript.
Point 10: Figure 4 caption: phylum should be phyla.
Reply: Thanks for the reviewer’s useful comments. We have corrected the mistake in the revised manuscript.
Point 11: Figure 5 caption: there are more than 10 families listed.
Reply: Thanks for the reviewer’s useful comments. What the figure showed are the 10 most abundant families in each group. As the top 10 families are different in different groups, the legend shows more than 10 families. For the misunderstanding caused by this, we have added an explanation to avoid misunderstanding.
Point 12: L 280: “of microbial” should be deleted.
Reply: Thank you for your precious comments and advice. We have revised the manuscript accordingly.
Point 13: Figure 6 caption: There are more than 10 genera listed.
Reply: Thanks for the reviewer’s useful comments. What the figure showed are the 10 most abundant genera in each group. As the top 10 genera are different in different groups, there are more than 10 genera listed in the Figure 6 caption. For the misunderstanding caused by this, we have added an explanation under the figure to avoid misunderstanding.
Point 14: L 308-309: One of these two comparisons should be HC and SC. Please correct.
Reply: Thank you for your precious comments and advice. We have revised the manuscript accordingly.
Point 15: Figure 7: Were all of these t-tests corrected for experiment-wise error so as not to inflate the Type I error rate? Simply by making so many tests, some will come up as significant by chance. Therefore the significance level (0.05) is often divided by the number of comparisons to avoid this problem. So for 10 comparisons, only those having a P-value less than 0.005 would be considered significant.
Reply: Thanks for the reviewer’s useful comments. After the literature review and discussion, we totally agree to correct the p-value. However, we think that the method of Benjaminiand Hochberg (BH) was more suitable. We have added this method to the ‘Materials and Methods’ and revised the manuscript accordingly.
Changes in manuscript:
L 196-202: Added “Finally, in order to avoid the occurrence of "Type I error", we corrected the p-value to q-value by Benjaminiand and Hochberg (BH) method as follows: 1) The p-values of each gene are ranked from the smallest to the largest; 2) The largest p-value remains as it is; 3) The second largest p-value is multiplied by the total number of genes in gene list di-vided by its rank. If less than 0.05, it is significant: q-value = p-value*(n/n-1); 4) The third p-value is multiplied as in step 3: q-value = p-value*(n/n-2); 5) And so on.”
About Fig 7: We finally showed results with p-value and q-value both less than 0.05. The results described in Figure 7 in the text are also modified accordingly.
L 326-335: Based on the t-test, it's possible to analyze the differences in gut microbiota between the three species of sea cucumber. At the phylum level, the number of phyla with differential gut microflora of three sea cucumber species is low and not statistically significant (Figure 7a). Specifically, the only significant differences in phylum are Acidobacteria (p <0.05, q <0.05) between SM and SC groups, and Schekmanbacteria (p <0.05, q <0.05) between SC and HA groups. The main differences in gut microbiota among the three species are at the family level (Figure 7b) and genus level (Figure 7c). At the family level, there were five families with differences between SM and HA (p <0.05, q<0.05). At the genus level, five genera are different between SM and HA (p <0.05, q<0.05).
Point 16: L 341: Should read “and did not completely overlap with each other”.
Reply: Thank you for your precious comments and advice. We have revised the manuscript accordingly.
Point 17: L 407: Whether oxygen is low in the sediment will depend on the depth at which it is sampled (which is why I asked above). In organic-rich sediments, the depth of the oxidized zone may only be a few mm.
Reply: We collected only sediment samples from the surface layer that was within 1 cm. The sediment in the adjacent area is not high-viscosity and density silt, but low-viscosity loose sand. Moreover, the sea cucumber samples were collected at a depth of 9-14 meters below the horizontal plane, which was still within the surface seawater with elevated dissolved oxygen levels. We referred to the data of dissolved oxygen in 10-meter seawater of this sea area during the same time period (DO 6.32 mg/L). In addition, studies have shown that the gut is a relatively hypoxic environment (Singhal et al., 2020; Cartwright et al., 2023). Therefore, we speculate that the oxygen content in the sediment is higher than that in the gut, and based on the experimental results we hypothesized that the higher oxygen content in the sediment than in the gut may be one of the reasons for the different microbial structures.
References:
Singhal, R.; Shah, Y. M. Oxygen battle in the gut: Hypoxia and hypoxia-inducible factors in metabolic and inflammatory responses in the intestine. J. Biol. Chem. 2020, 295, 10493-10505. Doi: 10.1074/jbc.REV120.011188
Cartwright, I. M.; Colgan, S. P. The hypoxic tissue microenvironment as a driver of mucosal inflammatory resolution.Front. Immunol. 2023,14, 1124774. Doi:10.3389/fimmu.2023.1124774
Point 18: L 459-460: This sentence is repeated twice. Please delete.
Reply: Thanks for the reviewer’s useful comments. We have corrected the mistake in the revised manuscript.
Point 19: L 483-485: Consider reworking the last sentence of the Conclusions for a more significant take-home message.
Reply Thanks for the advice of the experts, we have revised the grammar and order of the conclusion, so that it can be read more smoothly, and made corresponding modifications to the last sentence.
L 500-514: Here, we used a high-throughput 16S rRNA gene-based molecular microbiology approach to study the gut bacterial communities of three species of tropical sea cucumbers and sediment microflora. Lactobacillus and Pseudomonas were detected in the gut of the three sea cucumber species studied, suggesting they could be used as candidate probiotics. Future studies should evaluate the abundance and function of the potential probiotics in sea cucumbers. Based on KEGG metabolic pathway analysis, the gut microbiome of sea cucumbers plays a key role in gut metabolism, digestion, and absorption. We also observed distinct differences in the bacterial communities of the gut and sediment microflora of the three sea cucumber species studied. The unique gut environment likely contributes to these differences. Additionally, the gut microbiome also showed differences due to the different habitats of the three tropical sea cucumber species. It has been speculated that selective feeding may be the main reason for the different microbial communities in the gut microbiome of different sea cucumbers. The study sheds a microscopic light on the differences in the diets of different sea cucumbers and the relationship between gut and habitat microbes. These results provide important data for future research on the gut microbiota of tropical sea cucumbers.

Reviewer 2 Report
The manuscript presented by the Authors shows the diversity of gut microbiota found in three species of sea cucumbers and compared to environmental sediments. In order to characterize the structure of the microorganisms present in the intestines of the aforementioned invertebrates, the Authors of the manuscript used high-throughput 16S sequencing technology, each step of which is described in detail.The conducted studies indicate the presence of a lower abundance of intestinal microflora of sea cucumbers collected from the environment compared to surface sediment. The Authors of the study also highlight the presence of differences in the intestinal microflora found between the three species. I like the way the manuscript was prepared and I have included some, minor comments below.
Line 229: please correct OUT to OTU.
Figure 5, Figure 6: I suggest harmonizing the number of families and genera in the figures with the number given in the captions.
Line 458: " Previous studies showed that H. atra likes to feed on gravel and coarse sand..." this sentence is repeated twice .
Author Response
Response to Reviewer 2 Comments
Point 1: Line 229: please correct OUT to OTU.
Reply: Thanks for the reviewer’s useful comments. We have corrected the mistake in the revised manuscript, and it is an important reminder to us that we will examine our manuscripts more carefully in the future.
Point 2: Figure 5, Figure 6: I suggest harmonizing the number of families and genera in the figures with the number given in the captions.
Reply: Thanks for the reviewer’s useful comments. The relative abundance of the 10 most abundant families or genera in each set of samples is shown in each bar graph. However, since the top 10 bacteria at the family and genus levels are not exactly the same, the legend shows more than 10 bacteria. For the misunderstanding caused by this, we have added an explanation under the figure to avoid misunderstanding. What the figure showed are the 10 most abundant families(Figure 5) or genera(Figure 6) in each group. As the top 10 families or genera are different in different groups, there are more than 10 genera listed in Figure 6 caption.
Point 3: Line 458: " Previous studies showed that H. atra likes to feed on gravel and coarse sand..." this sentence is repeated twice.
Reply: Thanks for the reviewer’s useful comments. We have corrected the mistake in the revised manuscript.
